# LESS IS MORE: LEAN YET POWERFUL VISION-LANGUAGE MODEL FOR AUTONOMOUS DRIVING

## ABSTRACT

In this work, we reconceptualize autonomous driving as a generalized language and formulate the trajectory planning task as *next waypoint prediction*. We introduce Max-V1 [1], a novel framework for one-stage end-to-end autonomous driving. Our framework presents a single-pass generation paradigm that aligns with the inherent sequentiality of driving. This approach leverages the generative capacity of the VLM (Vision-Language Model) to enable end-to-end trajectory prediction directly from front-view camera input. The efficacy of this method is underpinned by a principled supervision strategy derived from statistical modeling. This provides a well-defined learning objective, which makes the framework highly amenable to master complex driving policies through imitation learning from large-scale expert demonstrations. Empirically, our method achieves the *state-of-the-art* performance on the nuScenes dataset, delivers an overall improvement of over 30% compared to prior baselines. Furthermore, it exhibits superior generalization performance on cross-domain datasets acquired from diverse vehicles, demonstrating notable potential for cross-vehicle robustness and adaptability. Due to these empirical strengths, this work introduces a model enabling fundamental driving behaviors, laying the foundation for the development of more capable *self-driving agents*. Code will be available upon publication.

## 1 INTRODUCTION

Human driving is an inherently sequential decision-making process, in which each action is conditioned on a real-time understanding of the surrounding scene. This dynamic interplay of perception and action exhibits strong similarities to natural language generation, which also involves producing a highly correlated sequence of outputs. Viewing the driving task from this perspective allows us to frame a Vision-Language Model (VLM) as a powerful policy network. In this context, the model's

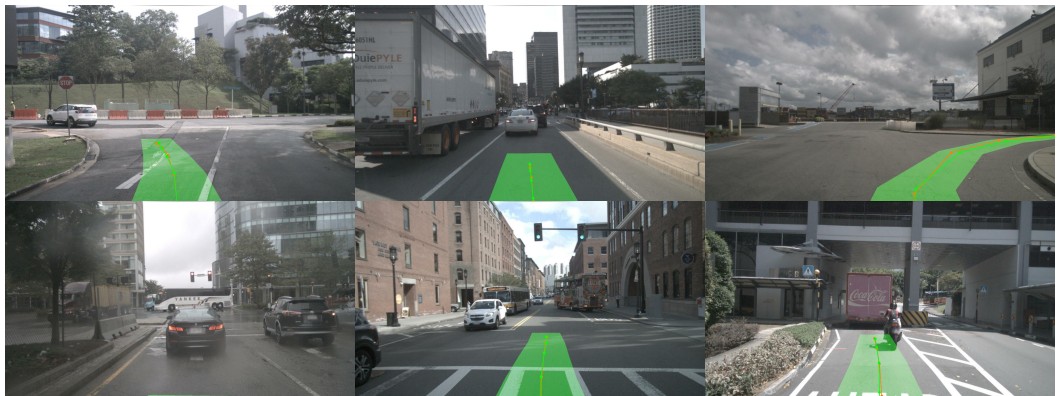

Figure 1: Visualization of typical driving scenarios. Predicted trajectories and ego vehicle coverage are shown in green, whereas ground truth trajectories are displayed in orange.

---

[1]In tribute to the renowned Dutch racing driver Max Verstappen.

objective shifts from predicting the next word to generating the next driving action, transforming the planning problem into a tractable, autoregressive sequence modeling task. This conceptual leap opens the door for leveraging the vast pre-trained knowledge and sophisticated reasoning capabilities of VLMs to tackle the complexities of autonomous driving.

The end-to-end approach has emerged as a dominant paradigm in autonomous driving, as it facilitates global optimization of the entire system and mitigates error accumulation. Within this paradigm, current research has diverged into two primary approaches. The first centers on developing bespoke architectures, trained exclusively on large-scale, domain-specific driving datasets. The second focuses on adapting large, pre-trained VLMs, aiming to leverage their vast world knowledge and general reasoning capabilities for the driving task.

The first approach, exemplified by methods like UniAD (Hu et al., 2023), typically employs carefully designed bespoke sequential architectures centered around Bird's-Eye View (BEV) representations. This approach is predicated on the assumption that a model, when meticulously trained on vast amounts of real-world driving data, can learn robust policies for practical deployment, with BEV serving as an efficient intermediate representation. However, this paradigm faces a notable dual challenge. On the one hand, its strong dependence on pattern recognition within high-quality curated datasets limits its generalization capabilities when encountering long-tail scenarios. On the other hand, the BEV representation itself introduces fragility: its generation from camera imagery is an ill-posed problem prone to information loss, and the scarcity of large-scale, accurately annotated BEV datasets remains a critical unavoidable bottleneck.

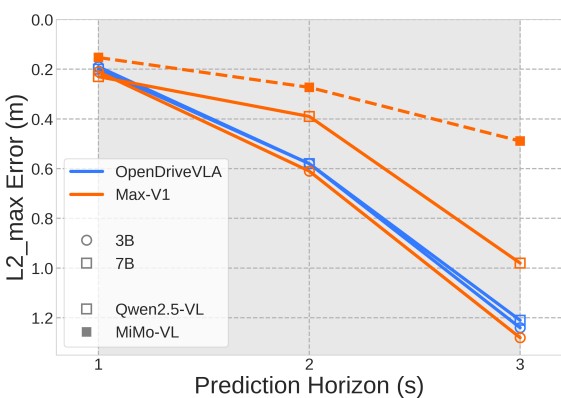

Figure 2: Comparison between our framework and OpenDriveVLA-3B/7B models (Zhou et al., 2025): both methods, which adopt `Qwen2.5-VL` as their base model, are denoted by solid lines.

The second approach, in contrast, more flexibly and effectively leverages mature VLM-related frameworks like those in (Jiang et al., 2024; Xing et al., 2024; Qiao et al., 2025) as high-level reasoning engines. By adopting Q&A format, these systems can deeply tap into and utilize the rich pre-trained knowledge of VLMs to further enhance their contextual awareness of driving scenarios. However, their generalist nature leads to a mismatch in autonomous driving task alignment: model architectures and objective functions optimized for discrete text processing are not naturally suited for the continuous, fine-grained control essential to real-world trajectory planning.

This analysis of current end-to-end approaches reveals two parallel schools of thought, each with inherent limitations. Specialized models are optimized for large-scale domain-specific datasets yet limited by their data-driven nature and fragile intermediate representations. The other focuses on VLM-related frameworks, which offer strong reasoning but face challenges with computational inefficiency and an inherent unsuitability for the continuous control problem. Developing more integrated architectures to bridge these gaps thus offers a promising evolutionary path and serves as the primary motivation for our work.

In this work, we present **Max-V1**, an end-to-end autonomous driving trajectory planner built on a pure VLM. Our approach enables a pre-trained VLM to acquire driving-related capabilities through fine-tuning solely on driving-specific behaviors, allowing the model to *focus on* the task. To achieve this, Max-V1 models driving as a sequential decision process similar to natural language and eliminates the traditional BEV feature space, instead processing raw sensor input directly from an egocentric, first-person perspective. By operating within this pure VLM-driven, end-to-end architecture, our paradigm combines both high performance and structural simplicity with the potential for robust cross-domain generalization. This approach avoids error accumulation from BEV construction, harnesses pre-trained knowledge, reduces dependency on costly BEV-specific annotations, and aligns more closely with the nature of driving. Specifically, we formulate our contributions as follows.

- We statistically model driving behavior as a sequential decision process and frame the planning task as *next waypoint prediction*, for which we demonstrate the validity of our supervision signal design. This formulation lays a principled foundation for our single-pass design and aligns with the nature of driving. We then leverage the pre-trained VLM as both a domain-specific knowledge repository and a powerful policy network to address this task via fine-tuning.

- Without any external information during training, our method achieves the *state-of-the-art* performance on the nuScenes dataset, delivers an overall improvement of over $30\%$ compared to prior baselines. In particular, our model demonstrates strong zero-shot generalization, exhibiting competent driving behavior in distinct scenarios. As these datasets were collected using entirely different vehicles, this performance indicates a strong potential for robust cross-vehicle deployment. In addition, we briefly explore first-person perspective LiDAR-image fusion, identifying a trade-off that leans more toward short-term objectives.

- Our framework provides a task-specific adaptation framework for VLMs to replace the conventional multi-stage driving pipeline. This unified architecture provides a structurally simplified foundation, making it a scalable foundation for the development of more capable *self-driving agents* through reinforcement learning.

## 2 RELATED WORKS

In this section, we review the literature across two key areas that inform our work. First, we examine the evolution of end-to-end autonomous driving. Second, we cover the recent integration of VLMs for high-level reasoning and control.

### 2.1 END-TO-END AUTONOMOUS DRIVING

Traditional autonomous driving systems typically employ a modular architecture, separating the pipeline into distinct perception, prediction, and planning stages, where each is trained independently with task-specific objectives. In contrast, end-to-end autonomous driving directly maps sensory inputs to planning outputs by uniting these stages under joint training, thereby minimizing cumulative information loss across multi-stage processing. UniAD (Hu et al., 2023) proposed a unified framework integrating core modules to enable comprehensive end-to-end planning optimization. As the first transformer-based system to cover such a complete set of driving tasks, it demonstrated the significant value of coordinated multi-task learning. The VAD (Jiang et al., 2023) and VADv2 (Chen et al., 2024) series further advanced this paradigm with vectorized scene representations, reducing the computational load while boosting the overall performance.

Conventional modular frameworks require perception modules to fully reconstruct the environment, and most end-to-end systems still follow this paradigm. However, emerging research showed that end-to-end architectures need only task-relevant perceptual features, an insight that has led to the navigation-guided implicit perception paradigm (Li & Cui, 2024), which uses driving-focused feature learning to improve inference efficiency.

### 2.2 VLMS FOR AUTONOMOUS DRIVING

Large models, with their strong reasoning, comprehension, and interpretability, can effectively address the limitations of end-to-end autonomous driving models. EMMA-style approaches represent a paradigm where an end-to-end VLM integrates visual inputs with natural language instructions, pioneering the adaptation of general VLMs to autonomous driving and demonstrating interpretability through driving-related reasoning. DriveGPT4 (Xu et al., 2024) processed front-camera video with VLMs to predict planning control signals and provide decision explanations. DriveVLM (Tian et al., 2024) used VLMs to predict coarse trajectories, and an end-to-end model further refined the generated trajectories. Senna (Jiang et al., 2024) proposed using VLMs to generate decisions and then leveraging these decisions to produce precise trajectory points, addressing the issue of insufficient numerical precision in large models. VERDI (Feng et al., 2025) distilled VLMs' reasoning and common sense into lightweight end-to-end models during training, eliminating reliance on VLMs at inference. Other studies (Sima et al., 2024; Qian et al., 2024) suggested using knowledge-augmented datasets to advance VLMs in autonomous driving.

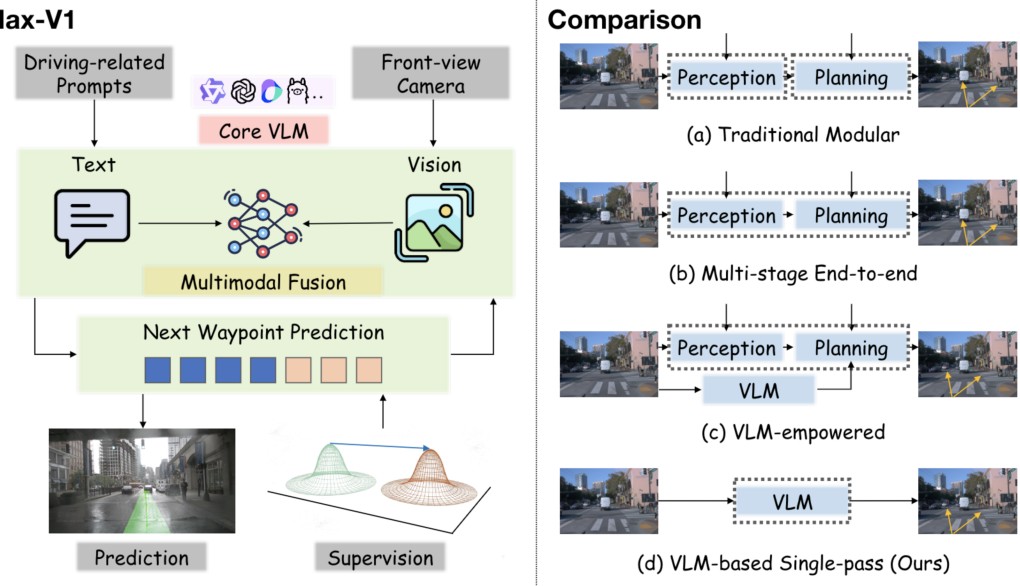

Figure 3: The architecture of our proposed method (Left). An overview comparing our method with mainstream paradigms is presented, in which dashed boxes denote independent end-to-end modules and arrows stand for additional input types required during training (Right).

## 3 METHODOLOGY

### 3.1 MODEL DESCRIPTION

#### 3.1.1 PRELIMINARY

GPT-style large language models (LLMs) operate on text sequences through an autoregressive mechanism. They are trained to predict the next token in a sequence given all preceding tokens, effectively learning the underlying probability distribution of the language. This is typically achieved by minimizing a cross-entropy loss, which enables the model to capture complex linguistic patterns.

When extended to multimodal contexts, these models, known as Vision-Language models (VLMs), are trained to generate a sequence of output tokens $\mathbf{O}$ based on a combination of textual prompts $\mathbf{T}$ and visual inputs $\mathbf{V}$. This process can be formally expressed as:

$$\mathbf{O} = \mathcal{M}(\mathbf{T}, \mathbf{V}). \tag{1}$$

where $\mathcal{M}$ represents the VLM. The generation remains autoregressive, with the model's output being a sequence of discrete semantic tokens.

#### 3.1.2 NEXT WAYPOINT PREDICTION

There exists a strong parallel between language generation and autonomous driving: both involve producing a highly correlated sequence of actions. Viewing the driving task from this perspective allows us to frame the VLM as a policy network, where the VLM's output yields the predicted trajectory, analogous to a sentence in language.

The primary challenge is therefore to represent this trajectory $\mathbf{W}_{\mathrm{BEV}}$ for a single sample as a sequence of waypoints $\mathbf{w}_t = (x_t, y_t)$, where $x_t$ and $y_t$ denote the coordinates of $\mathbf{w}_t$, within the autoregressive framework. A naive approach would be directly encoding waypoint coordinates into a textual format:

$$\mathbf{W}_{\mathrm{BEV}} = \{\mathbf{w}_t = (x_t, y_t)\} \rightarrow \texttt{text}\left(\{(x_t, y_t)\}\right), \tag{2}$$

Here, the textualized waypoints are treated as discrete tokens $\{\mathbf{s}_i\}_{i=1}^n$. Consequently, the model would be trained using the standard cross-entropy loss from LLMs:

$$\mathcal{L}_{\text{CE}} = -\sum_{j=1}^n \log\, p\left(\mathbf{s}_j \mid \mathbf{s}_0, \ldots, \mathbf{s}_{j-1}\right). \tag{3}$$

While the tokenization strategy is highly effective for natural language processing, it is poorly suited for autonomous driving. The core issue stems from a mismatch in data domains, as linguistic tokens are discrete semantic units, whereas waypoint coordinates are continuous values with direct physical meaning. Treating the latter as discrete words creates incompatibility with cross-entropy loss, and this incompatibility harms performance because cross-entropy, designed for categorical rather than continuous spatial data, fails to reflect geometric proximity. Thus, it penalizes minor waypoint deviations and completely erroneous locations equally, violating motion continuity and spatial metrics.

In contrast, a space-sensitive loss function directly resolves this mismatch. Instead of treating waypoints as discrete classes, it quantifies the geometric discrepancy between the predicted and ground-truth trajectories. By scaling the penalty according to the actual spatial deviation, the optimization process becomes better aligned with the physical requirements of smooth, continuous motion, which ultimately leads to superior performance.

To address this, we reframe *next word prediction* as *next waypoint prediction*, treating it as a regression problem within the autoregressive framework. We model the trajectory prediction using special tokens that serve as placeholders for continuous coordinate values:

$$\mathbf{W}_{\text{BEV}} = \{\mathbf{w}_t = (x_t, y_t)\} \rightarrow \texttt{tokenize}(\{(x_t, y_t)\}), \tag{4}$$

The model generates waypoints sequentially, preserving the autoregressive structure to capture temporal dependencies in motion:

$$p(\mathbf{w}_0, \mathbf{w}_1, \ldots, \mathbf{w}_T) = p(\mathbf{w}_0) \prod_{t=1}^T p(\mathbf{w}_t \mid \mathbf{w}_0, \mathbf{w}_1, \ldots, \mathbf{w}_{t-1}). \tag{5}$$

Unlike most of the LLMs whose cross-entropy loss is defined on discrete distributions for tokens, we model each waypoint, which corresponds to a token in the sequence, as a Gaussian distribution in the continuous space $\mathbb{R}^2$, namely,

$$p_t := p(\mathbf{w}_t \mid \mathbf{w}_0, \mathbf{w}_1, \ldots, \mathbf{w}_{t-1}) \sim \mathcal{N}(\boldsymbol{\mu}_t, \sigma^2 \mathbf{I}), \tag{6}$$

where $\sigma$ is a given constant for all $t$ and $\boldsymbol{\mu}_t$ is unknown, $p(\mathbf{w}_0)$ is omitted in the following context since the initial waypoint $\mathbf{w}_0$ is always set to be $[0.0, 0.0]^\top$, and then $p(\mathbf{w}_0) = 1$. Note that $\mathbf{w}_t$ is the only sample from $p_t$, so the maximum likelihood estimation of $p_t$ is

$$\tilde{p}_t \sim \mathcal{N}(\mathbf{w}_t, \sigma^2 \mathbf{I}). \tag{7}$$

Similarly, for the predicted waypoints $\mathbf{w}_t'$, the conditional distribution is defined as

$$q_t := p(\mathbf{w}_t' \mid \mathbf{w}_0', \mathbf{w}_1', \ldots, \mathbf{w}_{t-1}') \sim \mathcal{N}(\boldsymbol{\mu}_t', \sigma^2 \mathbf{I}), \tag{8}$$

and the maximum likelihood estimation is given as

$$\tilde{q}_t \sim \mathcal{N}(\mathbf{w}_t', \sigma^2 \mathbf{I}). \tag{9}$$

Then, the empirical cross-entropy loss for a single sample, defined between maximum likelihood estimation distributions $\tilde{p} := \prod_{t=1}^T p_t$ and $\tilde{q} := \prod_{t=1}^T q_t$, is given as

$$\tilde{\mathcal{L}}_{\text{CE}} = -\sum_{t=1}^T \log \tilde{q}_t(\mathbf{w}_t) \tag{10}$$

$$= \sum_{t=1}^T \frac{1}{2} \log\left(2\pi\sigma^2\right) + \frac{\|\mathbf{w}_t - \mathbf{w}_t'\|_2^2}{2\sigma^2}, \tag{11}$$

which, after neglecting constant terms, is equivalent to the $\ell_2$-loss

$$\mathcal{L} = \sum_{t=1}^{T} \|\mathbf{w}_t - \mathbf{w}'_t\|_2^2. \tag{12}$$

Crucially, instead of relying on cross-entropy loss for these special tokens, we introduce a task-specific loss tailored for waypoint regression. Consistent with physical intuition, we supervise the predicted coordinates against the ground truth using a physical distance loss:

$$\mathcal{L}_{\text{distance}} = \sum_{i=1}^{N} \sum_{t=1}^{T} \|\mathbf{w}_{i,t} - \mathbf{w}'_{i,t}\|_2^2, \tag{13}$$

where $\mathbf{w}_{i,t}$ represents the waypoint of sample $i$ at timestamp $t$. This approach offers two significant advantages over direct textual output.

- It resolves mismatch between the discrete nature of cross-entropy loss and the continuous nature of spatial data, while allowing for explicit control over numerical precision.
- By using compact special tokens instead of lengthy string representations, which fixes the output length for coordinates, significantly reduces token consumption and computational overhead during both training and inference process.

## 3.2 DISTINCTIONS FROM EXISTING WORKS

The emergence of VLM-based models, such as EMMA (Hwang et al., 2024; Zhou et al., 2025), has marked a milestone in planning for autonomous driving. Although our work shares a similar goal of leveraging VLM reasoning, it diverges in several critical design philosophies that are more foundational than the specific choice of base model. These distinctions are designed to optimize the directness and efficiency of *next waypoint prediction*, and their key differences are as follows:

- **Statistical Modeling.** Our approach distinguishes itself through the systematic understanding of supervision signals. Specifically, by conducting a thorough analysis of the inherent characteristics of driving task, we derive a statistically grounded model for waypoint representation. This provides a principled foundation for the proposed $\ell_2$-loss function. Compared to those based on the cross-entropy loss, the merits of our loss design are supported by intuitive reasoning, theoretical analysis, and empirical evidence. To the best of our knowledge, we are the first to carry out detailed theoretical modeling of the loss function itself in VLM-based driving research.
- **Single-Pass Generation.** A core design principle of our framework is its profound simplicity, achieved without reliance on auxiliary components such as additional Chain-of-Thought (Wei et al., 2022; Tian et al., 2024) annotations. This avoids the laborious and costly process of collecting detailed reasoning data. Our approach also eschews the multi-turn dialogue for iterative refinement. Instead, our framework is a single-pass, end-to-end methodology that generates the entire trajectory directly.
- **Lightweight Input.** Many existing methods depend on rich input modalities, among which ego-vehicle status stands out as a key component that provides substantial information, together with surround-view video and other forms of guidance information. In contrast, our approach is designed to operate solely on a single frame from a front-view camera, without requiring any additional ego-state information. Clearly, not only does our design improve the training and inference efficiency by reducing the input complexity, but it also better aligns with the human front-view driving intuition.

# 4 EXPERIMENTAL RESULTS

## 4.1 EXPERIMENT SETUPS

We summarize the key experimental setups here, with a more detailed description of our configurations and hyperparameters provided in Appendix E.

| Model | $L2_{avg}$ (m) $\downarrow$ | | | | $L2_{max}$ (m) $\downarrow$ | | | |
|---|---|---|---|---|---|---|---|---|
| | 1s | 2s | 3s | *Avg.* | 1s | 2s | 3s | *Avg.* |
| ST-P3 (Hu et al., 2022) | 1.33 | 2.11 | 2.90 | 2.11 | - | - | - | - |
| VAD (Jiang et al., 2023) | 0.41 | 0.70 | 1.05 | 0.72 | - | - | - | - |
| UniAD (Hu et al., 2023) | 0.42 | 0.64 | 0.91 | 0.66 | 0.48 | 0.96 | 1.65 | 1.03 |
| SparseDrive (Sun et al., 2024) | 0.29 | 0.58 | 0.96 | 0.61 | - | - | - | - |
| Senna (Jiang et al., 2024) | 0.26 | 0.42 | 0.61 | 0.43 | - | - | - | - |
| SSR (Li & Cui, 2024) | 0.19 | 0.36 | 0.62 | 0.39 | 0.25 | 0.64 | 1.33 | 0.74 |
| OpenDriveVLA (Zhou et al., 2025) | 0.15 | 0.31 | 0.55 | 0.33 | 0.20 | 0.58 | 1.21 | 0.66 |
| EMMA* (Hwang et al., 2024) | 0.14 | 0.29 | 0.54 | 0.32 | - | - | - | - |
| EMMA+* (Hwang et al., 2024) | **0.13** | 0.27 | 0.48 | 0.29 | - | - | - | - |
| **Max-V1**   `Qwen2.5-VL-3B` | 0.17 | 0.33 | 0.59 | 0.36 | 0.21 | 0.61 | 1.28 | 0.70 |
| `Qwen2.5-VL-7B` | 0.24 | 0.28 | 0.46 | 0.33 | 0.23 | 0.39 | 0.98 | 0.53 |
| `MiMo-VL-7B-SFT` | 0.24 | 0.38 | 0.65 | 0.42 | 0.28 | 0.63 | 1.41 | 0.77 |
| `MiMo-VL-7B-RL` | 0.15 | **0.20** | **0.27** | **0.21** | **0.15** | **0.27** | **0.49** | **0.30** |

*Note:* * denotes additional model details: EMMA is initialized from Google Gemini (Team, 2024); EMMA+ is pre-trained on Waymo's internal extra data.

Table 1: Main results. In this table, no additional information is used except sensor input. The $L2_{max}$ error is from UniAD and the $L2_{avg}$ error is from ST-P3. The *Avg.* column denotes the average of the first three seconds.

We strictly follow the official nuScenes dataset splits. In Table 1, all models are trained on `train` and evaluated on `val`. All other experiments use models trained on `trainval` for a more comprehensive assessment. Results on `test` are reported in Section B. For all subsequent experiments, we adopt the $L2_{max}$ error from UniAD as it is more intuitive than the $L2_{avg}$ error from ST-P3.

To mitigate data imbalance, we employ a balanced sampling strategy. Our model is designed to predict next ten waypoint tokens at $0.5s$ intervals. The model is guided by a specific prompt, and the full design of this prompt as well as its corresponding ablation studies are detailed in Appendix C. During testing, the model receives no ego-state information.

The model was trained on a computing node equipped with NVIDIA A100 (80GB) GPUs. To enhance robustness against error accumulation, we employ a curriculum learning strategy via *scheduled sampling*, which gradually exposes the model to its own predictions. The specific scheduling parameters and training hyperparameters are available in Appendix E.

## 4.2 MAIN RESULTS

To demonstrate the generality of our framework, we chose pre-trained VLMs from distinct research groups, including different versions of `Qwen2.5-VL` and `MiMo-VL` listed in Table 1. Under displacement error metrics, our Max-V1 framework demonstrates the *state-of-the-art* performance under distinct evaluation criteria.

Specifically, our Max-V1 framework establishes the *state-of-the-art* across both $L2_{avg}$ and $L2_{max}$ error, with the `MiMo-VL-7B-RL` variant leading the performance. For the average error, this model achieves the best result of $0.21m$, securing top performance at both the $2s$ and $3s$ horizons and substantially outperforming all non-VLM-based models. The same variant also exhibits superiority in $L2_{max}$ errors, with its average error of $0.30m$ being among the lowest and errors at each time step also remaining relatively low. Moreover, it is worth noting that other variants within the Max-V1 framework also demonstrate competitive performance. Our hypothesis is that the overall effectiveness of our framework derives from the combination of powerful base models and well-designed training procedures, which includes the selection of the supervisory signal.

The results mentioned above are restricted to the nuScenes dataset. Therefore, we believe that more out-of-distribution tests are needed to offer a more persuasive assessment. The zero-shot performance is particularly relevant, revealing the framework's real-world generalization. We validate its cross-domain robustness in unseen environments (e.g., UK and Netherlands) and its cross-vehicle

adaptability, which indicates potential for deployment across diverse platforms. Both capabilities are evidenced by the model's solid fundamental driving skills and effective speed adaptation in diverse scenarios. Detailed results, visualizations, and discussion are provided in Appendix A.

## 4.3 ABLATION STUDY

### 4.3.1 DIFFERENT TYPES OF SUPERVISION

In this section, we carried out experiments to examine the impact of replacing discrete tokens, which are commonly employed by VLMs, with continuous coordinate vectors for waypoint representation in autoregressive generation. Specifically, we converted vector-based labels into string formats and then worked with `Qwen2.5-VL-3B` directly, following the same approach as in normal LLMs.

| Type | $L2_{\max}$ (m) $\downarrow$ | | | |
|---|---|---|---|---|
| | 1s | 2s | 3s | *Avg.* |
| token | 1.58 | 3.12 | 5.01 | 3.24 |
| vector | **0.18** | **0.32** | **0.51** | **0.34** |

Table 2: Ablation study of different data type, both models use `Qwen2.5-VL-3B` as base model.

As shown in Table 2, using discrete tokens instead of continuous vectors for waypoint representation, degrades performance by nearly an order of magnitude for the 3B model, rendering this approach unusable for robust trajectory prediction. These results further confirm the efficacy of our proposed supervision.

Beyond poor metric performance, string-based format suffers from a critical failure mode: model hallucinations that produce structurally invalid outputs. In our evaluation, this resulted in a failure rate of 11.4%, where the generated text could not be parsed into valid coordinates. These parsing failures mainly caused by three types of errors:

- **Incomplete Trajectories.** Outputting fewer waypoints than prompted.
- **Malformed Waypoints.** Generating waypoints with incorrect dimensionality.
- **Invalid Characters.** Producing non-numeric text that cannot be converted to coordinates.

These formatting failures are an inherent byproduct of representing coordinates as text. The VLM's vast vocabulary presents a significant challenge for the constrained task of generating numerical waypoints. Consequently, even a well-trained model still has a non-negligible probability of sampling invalid characters, which reflects the model's instability, leading to structural errors that pose significant risks to driving safety. In contrast, our Max-V1 framework completely eliminates this failure mode by design, as it is structured to directly output $2D$ vectors, ensuring syntactically correct trajectories.

### 4.3.2 EXPLORATORY STUDY ON MULTI-SENSOR

As a brief exploration into multimodal fusion, we implemented a simple image-plane projection strategy on our lightest model, with full details provided in Section D. This approach is conceptually distinct from BEV-based methods like BEVFusion (Liu et al., 2023), and serves as a simpler baseline compared to prior image-plane work such as the PMF-series (Zhuang et al., 2021; Tan et al., 2024) and the more complex sequential refinement in PointPainting (Vora et al., 2020).

| Camera | LiDAR | $L2_{\max}$ (m) $\downarrow$ | | | |
|---|---|---|---|---|---|
| | | 1s | 2s | 3s | *Avg.* |
| ✓ | × | 0.18 | **0.32** | **0.51** | **0.34** |
| ✓ | ✓ | **0.16** | 0.55 | 1.37 | 0.68 |

Table 3: Ablation study of sensor configurations, with both models using `Qwen2.5-VL-3B` as base model.

As shown in Table 3, the fusion-based configuration improves short-term accuracy at the $1s$, while exhibiting increased error at the $2s$ and $3s$ time steps relative to the vision-only baseline. This reveals a clear trade-off, where short-term precision is gained at the expense of long-term stability.

We attribute this phenomenon to the dual nature of LiDAR data. The dense point cloud in the near field provides strong geometric information that the model successfully leverages for immediate

planning, validating the effectiveness of the first-person perspective fusion mechanism. However, the inherent sparsity of LiDAR data at greater distances creates a sharp drop-off in reliable geometric constraints. This nonuniform information density appears to create a *short-sighted* model, one that overrelies on near-field certainties and struggles to perform the robust vision-based extrapolation required for stable long-term prediction.

The trade-off we observed represents a key consideration for multi-sensor fusion. Although enhanced short-term precision is a valuable asset for planning systems with a high inference frequency, the degradation of long-term stability highlights a clear avenue for improvement. Future work could focus on developing more sophisticated fusion techniques, aiming to preserve near-field accuracy without sacrificing long-range planning.

## 5 Limitations and Future Work

In this section, we discuss the limitations of our current approach and outline several promising directions for future research.

- **Data Scaling and Diversity:** Training on additional open-loop real-world datasets like nuPlan (H. Caesar & Tan, 2021) and Waymo Open Dataset (Sun et al., 2020) may enhance the diversity of driving styles and the model's robustness, yet the value of incorporating unskilled driver data remains questionable.
- **Inference Efficiency:** Due to the inherent limitations of VLMs, which is a common issue across all VLM-based methods, inference latency remains a challenge for real-time deployment. Future directions include exploring efficient inference techniques, such as distillation and quantization, and pursuing hardware acceleration through the development of custom chips to boost inference speed.
- **Lack of Interpretability:** End-to-end *black-box* architecture inherently lacks direct interpretability. While this design choice prioritizes task performance and efficiency, we acknowledge the critical importance of explainability in autonomous driving. Future work may focus on developing hybrid architectures or post-hoc methods to bridge this gap.
- **Beyond Imitation Learning:** The current model is based on imitation learning, which cannot escape the limitations of expert demonstrations. The fine-tuning process could be enhanced by introducing reinforcement learning to allow the model to learn from interaction and discover more optimal driving policies.

## 6 Conclusion

In this work, we propose a novel framework termed **Max-V1** that adapts a general-purpose VLM for the task of trajectory planning in autonomous driving. Our approach is built upon a synergistic framework that integrates three core components: (i) a direct, autoregressive waypoint prediction policy; (ii) a task-specific fine-tuning strategy; and (iii) a concise, ego-centric input format. The planning process is guided by a statistically sound, physics-informed supervision. This method bypasses textual tokenization to align the model's predictions directly with driving behavior, resulting in robust end-to-end trajectory planning. Quantitatively, our model generally outperforms the previous *state-of-the-art* baseline in imitative performance: our displacement error metrics see an overall reduction of over $30\%$ across all evaluated trajectory planning items. This strong empirical performance, empowered by a key theoretical insight rooted in statistical modeling, underscores the practical viability of our approach. As a brief exploration, we have also undertaken a pilot study on a simple LiDAR fusion strategy, which reveals a clear performance trade-off and offers a novel direction for future enhancements.

Although standard displacement metrics in autonomous driving are known to favor imitation fidelity over real driving intelligence, our model achieves a level of performance that validates its core capabilities in driving, and visually, it even demonstrates more reasonable driving than human drivers in some scenarios. This achievement points to a critical direction for future work: boosting driving intelligence via reinforcement learning. In general, this work provides a solid foundation for pursuing both the efficiency and capability required for *self-driving agents*.

## LLM STATEMENT

LLMs are only used for paper polishing.

## REPRODUCIBILITY STATEMENT

All experiments were carried out on publicly available datasets following their official split, with pre-trained models sourced from the ModelScope platform and detailed settings provided in Section 4. Due to high computational costs, each experiment was run only once; to ensure fairness, our proposed method was directly trained and evaluated, while the performance of all baselines was adopted from their research papers, and both were assessed using identical mainstream metrics to avoid bias from inconsistent conditions and enable clear, reproducible performance comparison of our approach.

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

# A   ZERO-SHOT RESULTS

To robustly evaluate a model's generalization, a rigorous cross-domain methodology is employed to train on nuScenes (Boston and Singapore) and then conduct zero-shot testing on the completely unseen View-of-Delft (Palffy et al., 2022) (Delft) and Oxford RobotCar (Maddern et al., 2017) (Oxford) datasets. This approach deliberately introduces significant domain gaps to challenge the model's adaptability across diverse geographic and cultural contexts.

The model is tested against unique environmental features, such as the high density of cyclists in Delft's narrow streets and the challenging, variable weather and lighting conditions in Oxford. Such a demanding zero-shot evaluation ensures the model learns a fundamental, transferable representation of the physical world rather than merely memorizing regional traffic patterns.

|  | **nuScenes** | **nuScenes** | **View-of-Delft** | **RobotCar** |
|---|---|---|---|---|
| **Country** | USA | Singapore | Netherlands | UK |
| **Driving Side** | Right | Left | Right | Left |
| **City Type** | Historic city | Modern city | Historic town | Historic city |
| **Description** | Complex roads, Aggressive traffic | Dense roads, Orderly traffic | Narrow streets, Mixed traffic | Structured roads, Orderly traffic |
| **Participants** | Vehicles, Pedestrians | High pedestrian density | Dense cyclists, Pedestrians | Vehicles, Pedestrians |

Table 4: Comparison of geographic and traffic characteristics across the training (nuScenes) and zero-shot testing (View-of-Delft, Oxford RobotCar) datasets. Evaluation covering Boston and Singapore is detailed in our main experimental section.

For the VoD dataset, we utilize the entire dataset (including both `train` and `test` splits) for evaluation and do not perform any additional processing on the images.

For the Oxford RobotCar dataset, we select the `large_sample` segment, as it is in poor lighting conditions. Image processing is performed as follows: given the original $4:3$ aspect ratio and the consistent presence of the ego vehicle in the lower portion of the images, this section is cropped to adjust the effective aspect ratio to $16:9$.

The generation of waypoints adheres to an established methodology. First, global GPS coordinates, provided in UTM format, are read. Given the limited spatial range of the data, the influence of Earth's curvature is deemed negligible and thus disregarded. Based on the ego vehicle's global orientation, these UTM coordinates are then converted into ego-centric waypoints, which constitute the ground truth for trajectory prediction.

Furthermore, due to domain shifts arising from different geographical regions, a common challenge emerges: the model may predict a geometrically plausible trajectory, yet its speed profile is inconsistent with the ground truth.

To address this issue, we employ a *coarse approximation* approach by introducing an optimal speed scaling factor $\lambda^*$, which serves as a simple yet effective mechanism to correct global speed discrepancies and enable disentangled evaluation of the trajectory's geometric accuracy against its speed profile. Moreover, the value of $|\lambda^* - 1|$ serves as an indicator of the model's cross-domain speed adaptation. A value closer to $0$ suggests better performance, as it signifies the model's ability to appropriately adjust its driving speed to the new scenario without significant rescaling.

Formally, given a predicted trajectory $\hat{\mathbf{W}} = \{\hat{\mathbf{w}}_i\}_{i=1}^N$ and its corresponding ground truth $\mathbf{W} = \{\mathbf{w}_i\}_{i=1}^N$, we find the optimal scaling factor $\lambda^*$ by solving the following least-squares problem:

$$\lambda^* = \arg\min_{\lambda} \sum_{i=1}^N \|\lambda \hat{\mathbf{w}}_i - \mathbf{w}_i\|_2 \,. \tag{14}$$

After obtaining $\lambda^*$, the rescaled trajectory $\lambda^* \hat{\mathbf{W}}$ can be used for a supplementary evaluation. This provides a clearer assessment of the model's capability to predict the geometric path, correcting for global errors in speed estimation.

## A.1 DELFT

| Model | | $L2_{\max}$ (m) $\downarrow$ | | | | $L2_{\max}$ (m) $\downarrow$ | | | |
| --- | --- | --- | --- | --- | --- | --- | --- | --- | --- |
| | | $\lambda$ | 1s | 2s | 3s | *Avg.* | $\lambda^*$ | 1s | 2s | 3s | *Avg.* |
| **Max-V1** | Qwen2.5-VL-3B | 1.0 | 0.47 | 0.83 | 1.27 | 0.86 | 1.01 | 0.47 | 0.82 | 1.27 | 0.85 |
| | Qwen2.5-VL-7B | 1.0 | 0.64 | 0.79 | 1.10 | 0.85 | 0.99 | 0.66 | 0.80 | 1.06 | 0.84 |
| | MiMo-VL-7B-SFT | 1.0 | 0.41 | 0.48 | 0.92 | 0.60 | 0.96 | 0.47 | 0.49 | 0.75 | 0.57 |

Table 5: Zero-shot performance in Dutch driving environments is evaluated via zero-shot testing on the View-of-Delft dataset. The table compares the $L2_{\max}$ error of raw ($\lambda = 1.0$) versus optimally rescaled ($\lambda^*$) predictions.

To rigorously evaluate zero-shot generalization under a significant domain shift, we selected VoD dataset. Collected entirely within a single European town, its scenarios often feature narrow streets, sometimes lacking clear road markings, and a more complex mixture of traffic participants. This environment contrasts sharply with the urban settings of our training data. For this evaluation, we conduct a comprehensive test across the entire VoD dataset, employing the checkpoints that demonstrated the best performance on the nuScenes in-domain testing. The results of this cross-domain transfer are presented in Table 5.

Among the evaluated approaches, our method based on `Mi-Mo-VL-7B-SFT` achieves the strongest overall performance, as indicated by its superior $L2_{\max}$ error in Table 5. This demonstrates its excellent capability to generalize the geometric path of the trajectory to a novel domain. However, a more nuanced analysis of the speed scaling factor, $\lambda^*$, reveals an intriguing trade-off. While the other models yield $\lambda^*$ values closer to the ideal 1.0, the best-performing model, namely `MiMo-VL-7B-SFT`, produces the lowest $\lambda^*$ value.

Although our speed scaling factor modeling is relatively coarse, this discrepancy suggests a compelling hypothesis: the `MiMo-VL-7B-SFT` model may be achieving its superior path accuracy by adopting a more aggressive, high-speed driving policy learned from the US and Singaporean training data. This strategy, while geometrically effective, is less appropriate for the narrow streets of the European town, forcing the post-hoc optimization to significantly scale down its speed. This finding highlights a critical challenge for future work: disentangling the learning of geometric paths from the adaptation of speed profiles to ensure that high trajectory accuracy does not come at the cost of unsafe or contextually inappropriate speed.

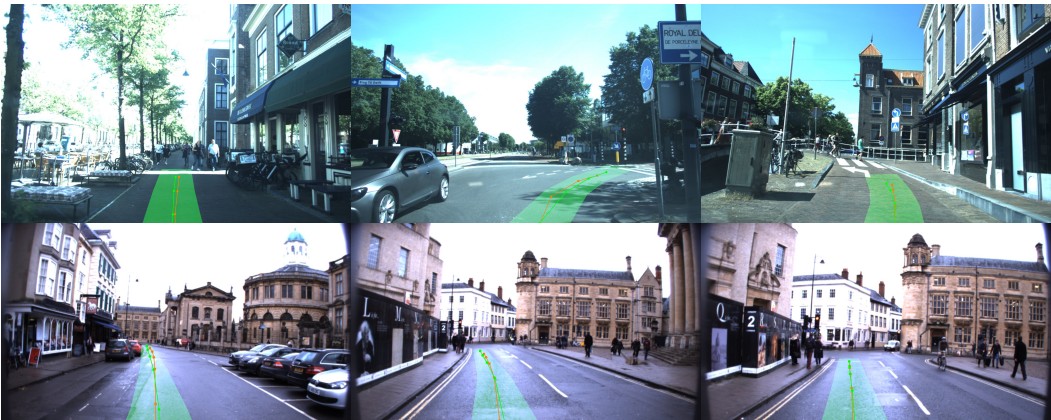

Figure 4: Visualization of frames from Delft (Top) and Oxford (Bottom). Predicted trajectories and ego vehicle coverage are shown in green, whereas ground truth trajectories are displayed in orange.

## A.2 OXFORD

| Model | | $L2_{\max}$ (m) $\downarrow$ | | | | $L2_{\max}$ (m) $\downarrow$ | | | | |
|---|---|---|---|---|---|---|---|---|---|---|
| | | $\lambda$ | 1s | 2s | 3s | *Avg.* | $\lambda^*$ | 1s | 2s | 3s | *Avg.* |
| **Max-V1** | Qwen2.5-VL-3B | 1.0 | 0.49 | 0.35 | 0.79 | 0.54 | 1.01 | 0.43 | 0.21 | 0.48 | 0.37 |
| | Qwen2.5-VL-7B | 1.0 | 0.18 | 0.60 | 1.47 | 0.75 | 0.93 | 0.43 | 0.21 | 0.48 | 0.38 |
| | MiMo-VL-7B-SFT | 1.0 | 0.65 | 0.65 | 0.45 | 0.58 | 1.02 | 0.56 | 0.49 | 0.65 | 0.56 |

Table 6: Generalization performance in UK driving environments is evaluated via zero-shot testing on the Oxford RobotCar dataset. The table compares the $L2_{\max}$ error of raw ($\lambda = 1.0$) versus optimally rescaled ($\lambda^*$) predictions.

To evaluate the zero-shot generalization of our models further, we then test them on standard clips from Oxford dataset. These scenes feature clear road structures and involve fundamental driving scenarios such as driving forward and stopping. The models are trained exclusively on the nuScenes dataset, and for this evaluation, we select several checkpoints that demonstrate the best performance on the nuScenes in-domain testing. The results of this cross-domain transfer test are presented in Table 6.

As shown in the table, the method based on Qwen2.5-VL-3B achieves the lowest overall $L2_{\max}$ error. Furthermore, its excellent performance is complemented by the optimal speed scaling factor $\lambda^*$ that remains close to the ideal value of $1.0$. This indicates that the model not only predicts geometrically accurate trajectories but also maintains an appropriate speed profile in a completely unseen environment, showcasing strong domain transfer capabilities. In contrast, the two 7B-parameter models, despite achieving superior results on the in-domain nuScenes benchmark, exhibit a relative performance degradation compared to the 3B-parameter model in this zero-shot scenario. We hypothesize that this is a consequence of the larger models overfitting to the nuances of the training domain, which harms their ability to generalize to novel environments.

## A.3 SUMMARY

Our zero-shot evaluations yield several key insights into the model's performance and robustness.

- Concerning model selection, MiMo-VL-7B-SFT consistently demonstrates the strongest performance across the evaluated out-of-distribution datasets, followed by Qwen2.5-VL-3B. This suggests a potential correlation between model scale and performance in complex, novel scenarios, although we hypothesize this advantage may diminish in simpler road conditions where smaller models might suffice.

- Regarding robustness, the results indicate that our method is remarkably resilient to variations in sensor parameters, such as camera intrinsics and extrinsics. Given that the training and zero-shot datasets originate from entirely different vehicle platforms and sensor rigs, this characteristic provides strong evidence for the model's potential for robust cross-vehicle deployment.

# B MORE RESULTS

In addition to our main results, we also report performance evaluated on the official nuScenes test split, with models trained on the combined trainval split. It is crucial to clarify the validity of this evaluation protocol. Notably, while the test split withholds $3D$ object bounding boxes and high-level scene descriptions to ensure fair evaluation via the official server, it does provide all the necessary information, including GPS data and ego-vehicle pose, to generate ground-truth trajectories, detailed prompt design is shown in C. This allows for a direct and legitimate assessment of planning performance.

The results, presented in Table 7, show a general improvement in performance compared to our primary experiments. We hypothesize this enhancement stems from two potential factors.

| Model | | $L2_{\max}$ (m) $\downarrow$ | | | | |
|---|---|---|---|---|---|---|
| | | 1s | 2s | 3s | *Avg.* | 5s |
| **Max-V1** | Qwen2.5-VL-3B | 0.18 | 0.32 | 0.51 | 0.34 | 0.66 |
| | Qwen2.5-VL-7B | 0.15 | 0.28 | 0.40 | 0.27 | 0.76 |
| | MiMo-VL-7B-SFT | **0.11** | **0.23** | 0.45 | **0.26** | 0.66 |
| | MiMo-VL-7B-RL | 0.21 | 0.28 | **0.32** | 0.27 | **0.41** |

Table 7: Results on the `test` split, with models trained on the `trainval` split. The *Avg.* column denotes the average of the first three seconds.

- Incorporating the `val` split into training provides the model with a richer and more diverse set of driving scenarios.

- It is possible that the distribution of scenarios within the `test` split is inherently less complex than that of the `val` split on which our main results are based.

Regardless of the precise cause, these results confirm the strong performance of our model under the official training and testing protocol.

## C  QUESTION DESIGN

The complete driving-related question prompts are structured as follows:

```
You are given a description of the current scene:
{scene description}.  You want {instruction}.  You
are a responsible driver, you need to follow the rules
of the road and stay safe as efficiently as possible.
Every 0.5s, the coordinates are represented by [x,
y], where x is the front and y is the left and right
direction, and the trajectory of the future 5s is
output in the format [x1, y1], [x2, y2],..., [x10,
y10]].
```

The boldface parts, namely **scene description** and **instruction**, refer to corresponding pieces of detailed context. We will further elaborate on the source of this text.

### C.1  SCENE DESCRIPTION

The `scene description` component is derived from the official nuScenes annotations.

During the training phase, each scene is accompanied by a brief textual summary. Provided by the dataset creators, these summaries are an inherent component of the dataset, and examples of such summaries include "`Construction, maneuver between several trucks`" and "`Intersection, peds, waiting vehicle, parked motorcycle at parking lot`", both detail the events unfolding in the scene. We directly inherit these scene descriptions from the dataset, and notably, all samples from the same scene share this identical description.

This flexibility in providing scene descriptions applies to our evaluations on the `validation` split, where we can leverage the available dataset annotations to construct informative prompts. However, the protocol for the official `test` split is necessarily stricter to ensure fair and unbiased evaluation. The `test` split does not provide any scene descriptions. Adhering to this principle, we do not inject any external information at this stage. Instead, for every scene in the `test` set, we consistently use the placeholder text "`## No descriptions available for the test set. ##`" as the prompt's descriptive component.

## C.2 INSTRUCTION

The `instruction` component of our prompt, used exclusively during the training phase, is sourced from the doScenes dataset (Roy et al., 2024). These instructions specify the high-level maneuver the ego-vehicle should execute, enabling the model to learn the association between linguistic commands and driving behaviors. Examples include "`follow the road`" or "`follow the car ahead...`". For a comprehensive overview of the instruction generation process, we refer the reader to the original paper.

However, during all evaluation phases, this component is intentionally left empty. This ensures a fair and challenging assessment of the model's planning capabilities, as it should generate trajectories based on visual input without any high-level guidance. This protocol prevents the introduction of external or manually crafted information during testing.

## C.3 ABLATION STUDY OF HIGH-LEVEL SCENE DESCRIPTIONS

To investigate the influence of high-level guidance on our model's decision making, we conduct an ablation study on the scene description component of the prompt. We compare the performance of our `Qwen2.5-VL-7B` based model under two conditions: one with scene descriptions provided and one without, where the description field is left empty. This test is conducted in the `val` split.

| Scene Description | $L2_{\text{avg}}$ (m) $\downarrow$ | | | | $L2_{\text{max}}$ (m) $\downarrow$ | | | |
|:---:|:---:|:---:|:---:|:---:|:---:|:---:|:---:|:---:|
| | 1s | 2s | 3s | *Avg.* | 1s | 2s | 3s | *Avg.* |
| ✓ | 0.24 | 0.28 | 0.46 | 0.33 | 0.23 | 0.39 | 0.98 | 0.53 |
| ✗ | 0.25 | 0.29 | 0.46 | 0.33 | 0.23 | 0.38 | 0.97 | 0.53 |

Table 8: Ablation study on the effect of scene descriptions, based on `Qwen2.5-VL-7B`.

As shown in the ablation results, the inclusion of high-level scene descriptions has a negligible impact on the model's performance. We hypothesize that this robustness to the presence or absence of explicit instructions is derived from several complementary factors.

- **Pre-trained World Knowledge:** The knowledge encoded within the VLM already serves as a vast repository of driving-related information. This forms a strong *scene-level prior*, enabling the model to make reasonable decisions in common scenarios without needing explicit textual guidance.

- **Richness of Visual Input:** The front-view image provides a rich and comprehensive perception of the immediate environment. Consequently, scene descriptions often become redundant, merely verbalizing information that is already salient and fully captured in the visual input.

- **Implicit Supervision from Trajectories:** The training data itself, with its trajectory labels, already embeds the necessary guidance. The model probably learns to infer the *intended trajectory* directly from the visual context, supervised by the associated ground truth. This includes not only the ego-vehicle's path but also anticipating the maneuvers of other traffic participants, thereby developing an implicit understanding that overlaps with the function of an explicit instruction.

- **Irrelevance of Static Instructions:** The static, scene-level descriptions lack the temporal granularity required for high-frequency decision-making. A single instruction covering an entire scene cannot adapt to sudden, dynamic events. Consequently, the model may end up assigning less weight to these static instructions, in preference to more immediate visual cues.

Ultimately, these findings suggest our framework learns a robust mapping directly from visual perception to driving actions via imitation learning. The model's strong imitation capabilities, combined with its powerful pre-trained priors, prove sufficient to generate competent driving behavior, rendering explicit, high-level instructions largely redundant in the tested scenarios.

## D  PROJECTION AND DEPTH NORMALIZATION

The prevailing paradigm in multi-stage end-to-end systems centers on BEV representations, a common medium for fusing sensor data like LiDAR. Yet, this reliance on an artificial construct inevitably leads to information loss and computational inefficiency.

We pivot away from this approach by utilizing VLMs, which are pre-trained on vast image datasets and naturally operate from a first-person perspective akin to human vision. However, we recognize that VLMs are ill-equipped to process dense video for dynamic spatial awareness, a critical gap that LiDAR's precise geometric data can fill.

Consequently, instead of forcing a VLM to adopt the unnatural BEV space, our fusion strategy is far more direct: we project LiDAR point clouds into the VLM's native first-person view. This allows for a seamless integration of semantic (image) and geometric (LiDAR) information, maximizing the strengths of both modalities within a unified and efficient framework.

For each $3D$ point $\mathbf{P}_i = (x_i, y_i, z_i)^\top$ in the point cloud, we first represent it in homogeneous coordinates by appending a fourth component: $\mathbf{P}'_i = (x_i, y_i, z_i, 1)^\top$.

The point is then transformed into the camera's coordinate system to obtain the projected point $\tilde{\mathbf{P}}_i = (\tilde{x}_i, \tilde{y}_i, \tilde{z}_i)^\top$. This is computed as:

$$\tilde{\mathbf{P}}_i = \mathbf{TRP}'_i,$$

where $\mathbf{R} \in \mathbb{R}^{4\times4}$ is a rigid transformation matrix (e.g., rotation and translation) and $\mathbf{T} \in \mathbb{R}^{3\times4}$ is the camera intrinsic matrix that projects points into the image plane.

The corresponding pixel coordinates $(u_i, v_i)$ in the image plane are then calculated via perspective division:

$$u_i = \frac{\tilde{x}_i}{\tilde{z}_i}, \quad v_i = \frac{\tilde{y}_i}{\tilde{z}_i},$$

The depth value $d_i$ for each point $\mathbf{P}_i$ is defined as its Euclidean distance from the origin of the LiDAR sensor's coordinate system:

$$d_i = \sqrt{x_i^2 + y_i^2 + z_i^2},$$

A depth map $\mathbf{D}$ is constructed from these projected points. The depth values are clamped to a predefined maximum value, denoted by $d_{\max}$. If multiple LiDAR points project to the same pixel, the one with the smallest depth value is used, effectively handling occlusions. For pixels that have no corresponding LiDAR points, the depth value is set to 0.

The value of the depth map $\mathbf{D}$ at pixel $(u, v)$ is formally defined as:

$$\mathbf{D}(u,v) = \begin{cases} \min\left(\min_{i \in S_{(u,v)}}\{d_i\}, d_{\max}\right), & \text{if } S_{(u,v)} \neq \emptyset \\ 0, & \text{otherwise} \end{cases}$$

where $S_{(u,v)}$ is the set of indices of all points $\{\mathbf{P}_i\}$ that project onto the pixel coordinates $(u, v)$, and $d_{\max}$ is the maximum depth threshold.

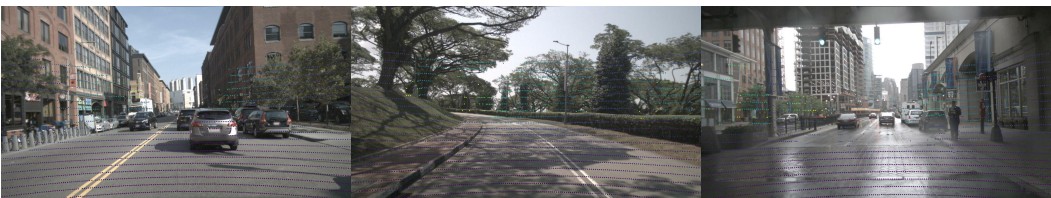

Figure 5: Visualizations of projected LiDAR point clouds.

Finally, this generated depth map is concatenated with the corresponding RGB image to form the RGB-D image, which serves as input to the large model. Detailed pseudocode is included in the following.

---

**Algorithm 1** Generation of RGB-D Images

---

**Input**: Image $I$, Point Clouds $\{\mathbf{P}_i\}$, Maximum Depth Threshold $d_{\max}$
**Output**: RGB-D Image $I_{\text{RGB-D}}$

1: Let $[H, W] \leftarrow$ `I.size()` (where $H$ is image height, $W$ is image width).
2: Initialize depth map $\mathbf{D} \leftarrow$ `zeros`$(H, W)$.
3: **for** each point $\mathbf{P}_i$ in $\{\mathbf{P}_i\}$ **do**
4:     Compute projected coordinates via perspective division and get pixel indices: $(u, v)$
5:     **if** $0 \leq u < W$ and $0 \leq v < H$ **then**
6:         Calculate depth value: $d_i$
7:         **if** $\mathbf{D}(u, v) = 0$ **then**
8:             $\mathbf{D}(u, v) \leftarrow \min(d_i, d_{\max})$
9:         **else**
10:            $\mathbf{D}(u, v) \leftarrow \min(\mathbf{D}(u, v), d_i, d_{\max})$
11:         **end if**
12:     **end if**
13: **end for**
14: Concatenate images: $I_{\text{RGB-D}} \leftarrow$ `Concat`$(I, \mathbf{D})$.
15: **return** $I_{\text{RGB-D}}$

---

During training, for the vision encoder, the weights of RGB channels are directly inherited from pre-trained model to preserve information, and the weights of the depth channel are Xavier-initialized (Glorot & Bengio, 2010).

# E  DETAILED EXPERIMENT SETTING

## E.1  BASIC SETUPS

This section provides a comprehensive overview of the configurations and conventions used in our experiments, complementing the summary in the main paper.

**Coordinate System.** To train and test our model, we follow the official split of the mainstream nuScenes dataset. All spatial data, including our waypoint predictions, are defined within the ego-vehicle coordinate system, as illustrated in Figure 6. In the nuScenes dataset, the ego-vehicle frame is conventionally defined by the raw LiDAR sensor's coordinate system, where the positive $x$-axis points forward, the positive $y$-axis points to the left, and the positive $z$-axis points upward.

**Note on Loss Function.** While our loss function, presented in Equation 13, allows for the implementation of a time-varying weight decay coefficient to sharpen the model's focus on short-term predictions. We choose not to utilize this method in our current implementation, to maintain simplicity and establish a strong but clear baseline.

**Data Sampling Strategy.** To address the inherent data imbalance in driving scenarios, we adopt a balanced sampling strategy during training. The dataset is categorized based on the vehicle's simple heading direction: *driving straight*, *turning left*, *turning right*, and *waiting*. The probability $p$ of selecting any given sample is set to $1/n_i$, where $n_i$ represents the total number of samples in that sample's heading category.

**Computational Resources.** The model was trained for approximately five days on a computing cluster equipped with $8\times$ NVIDIA A100 (80GB) GPUs.

## E.2  SCHEDULED SAMPLING DESCRIPTION AND SETUP

*Exposure bias* (Bengio et al., 2015) is a key challenge for autoregressive models that the training performance does not transfer to inference. This stems from a fundamental discrepancy between the training and inference procedures.

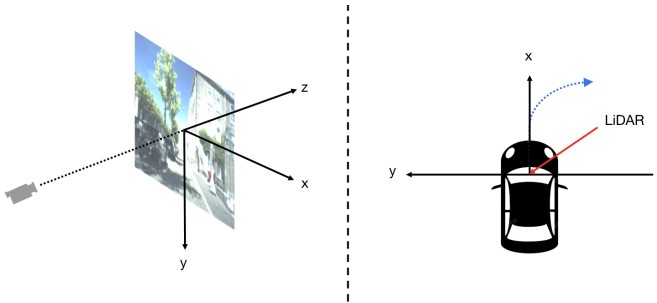

Figure 6: Camera and BEV coordinates

During training, under a teacher-forcing regime, the model predicts the $k$-th waypoint $\hat{\mathbf{w}}_k$ conditioned on the ground-truth history:

$$\hat{\mathbf{w}}_k = \mathcal{M}(\mathbf{T}, \mathbf{V}, \mathbf{w}_0, \ldots, \mathbf{w}_{k-1}). \tag{15}$$

In contrast, during inference, the model must rely on its own generated waypoints $(\hat{\mathbf{w}}_0, \ldots, \hat{\mathbf{w}}_{k-1})$, as the ground truth is unavailable. This mismatch causes minor deviations to compound and leads to wrong trajectories.

*Scheduled sampling* is a common technique to mitigate exposure bias. At each training step, the input for the next waypoint is stochastically selected between the ground-truth and the model's own previous prediction:

$$\mathbf{w}'_{k-1} = \begin{cases} \hat{\mathbf{w}}_{k-1}, & \text{with probability } p_{ss} \\ \mathbf{w}_{k-1}, & \text{with probability } 1 - p_{ss} \end{cases} \tag{16}$$

The model is then conditioned on this mixed history,

$$\hat{\mathbf{w}}_k = \mathcal{M}(\mathbf{T}, \mathbf{V}, \mathbf{w}'_0, \ldots, \mathbf{w}'_{k-1}). \tag{17}$$

Forcing it to learn a robust policy that can recover from imperfect inputs.

In our implementation, we progressively increase the probability $p_{ss}$ over the course of training, effectively hardening the curriculum. Specifically, we initialize the sampling probability at $0.4$ and linearly increase it by $0.1$ after each training epoch, capping it at a final value of $0.6$.

This particular scheduling strategy is designed with a clear purpose. The initial phase with lower $p_{ss}$ allows the model to first learn the basic driving patterns under strong supervision, while the gradual increase increasingly challenges the model to correct its own mistakes. Capping the probability at $0.6$ serves to stabilize the training process, ensuring that a consistent ground-truth signal is present to prevent potential divergence. Ultimately, this strategy of progressively forcing the model to become self-reliant is crucial for mitigating error accumulation and significantly improving trajectory stability during inference.

# F   MORE VISUALIZATIONS

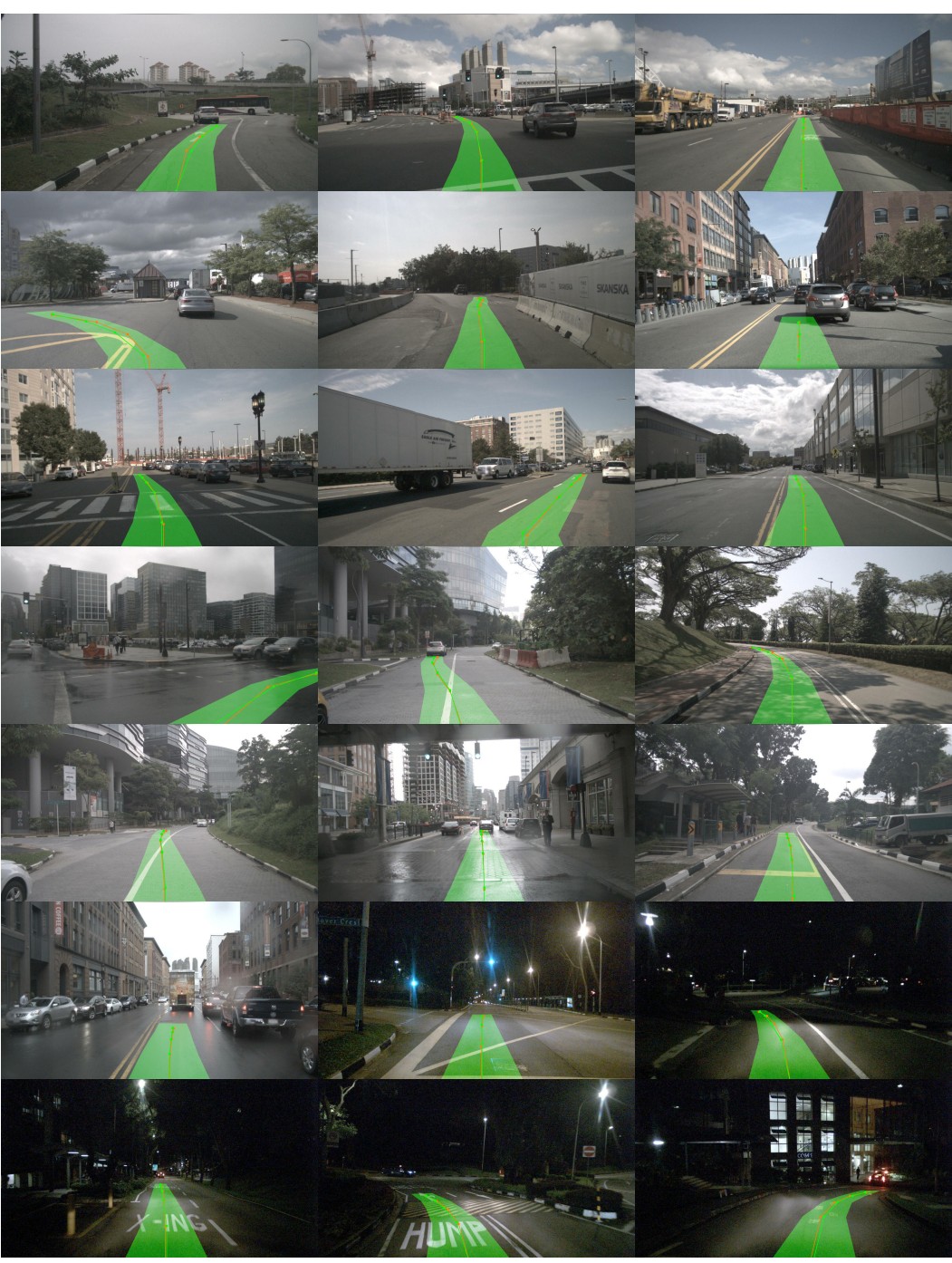

Figure 7: We uniformly sampled more frames from the nuScenes and visualized them. These results are generated using one of the top-performing model checkpoints, with predicted trajectories in green and ground truth in orange.

