# OpenReview forum: "Less is More: Lean yet Powerful Vision-Language Model for Autonomous Driving"
_ICLR.cc/2026/Conference — ICLR 2026 Conference Withdrawn Submission_

### Official Review · Reviewer_AtG6 · 2025-10-15

**Soundness:** 2
**Presentation:** 2
**Contribution:** 1
**Rating:** 2
**Confidence:** 5

**Summary:**

The paper introduces a new method to predict the future waypoint using VLM. Its main contribution lies in replacing the discretized token with continuous coordinate for the output. Its prediction performance is good on the nuScenes dataset, surpassin the previous VLM and E2E works.

**Strengths:**

1. The paper conducts out-of-the-distribution experiment to demonstrate its zero-shot performance.
2. The prediction accurancy is high in the nuScenes dataset.
3. The supplementary material is detailed for better understanding.

**Weaknesses:**

1. Replacing the tokenized space with the continous coordinate is not novel in E2E autonomous driving.
2. The paper only compare the L2 performance on the simple nuScenes dataset. The L2 performance is weakly coorelated with the final performance which has been widely expored in previous works because of the accumulation errors and multi-modality. Considering evaluating on more benchmarks such as Navsim and Bench2Drive, CARLA v2 with more comprehensive driving-related metrics.
3. The training detail is missed such  as the learning rate, learning epoch.
4. The figure 2 is not easy to understand. Consider replacing it with a histgram.

**Questions:**

1. Why is the token result in the ablation study much worse than the result of OpenDriveVLA which also uses token?
2. Why does the paper not report the collision result on the nuScenes dataset.
3. Why does MiMo-VL-7B-SFT get smaller error with longer prediction time in Table 6?
4. Why does the zero-shot experiment ignore the MiMo-VL-7B-RL model.
5. Why does the MiMo-VL-7B-RL achive much better performance on the validation but not on the test set?

---

### Official Review · Reviewer_a5V2 · 2025-10-31

**Soundness:** 2
**Presentation:** 3
**Contribution:** 1
**Rating:** 2
**Confidence:** 5

**Summary:**

This paper introduces Max-V1, a single-stage, end-to-end autonomous driving framework that formulates trajectory planning as an autoregressive next-waypoint prediction task. By leveraging a Vision-Language Model (VLM) and a statistically-derived $l_2$ loss, the model directly predicts continuous waypoints from a single front-view camera image, achieving state-of-the-art results on the nuScenes dataset.

**Strengths:**

1. Strong Empirical Performance: The method achieves state-of-the-art performance on the nuScenes dataset, outperforming prior baselines. It also shows promising zero-shot generalization to unseen datasets.
2. Deep Analysis of Results: The paper provides a thorough analysis of its results, particularly in the zero-shot generalization experiments. The discussion on trade-offs between model size, geometric accuracy, and speed adaptation in unseen domains provides valuable insights into the model's behavior.

**Weaknesses:**

1. Lack of Novelty: The framework of using a VLM to predict an action sequence is known as Vision-Language-Action (VLA) model. The paper lacks a sufficient literature review of existing VLA models for driving and does not clearly articulate its novelty or key differences from these works. The authors should better justify the advantages of this specific architecture over other VLAs.

2. Insufficient Ablation Study: The paper is missing critical ablation studies to justify its design choices. For example: a) what is the performance if all waypoints are predicted in parallel (as a whole set) instead of autoregressively? b) what is the performance of a standard pre-trained vision transformer (e.g., ViT + transformer decoder) trained with the same $l_2$ loss, to isolate the benefit of the VLM's pre-trained weights? The ablation on removing prompts (Table 8) shows a negligible impact, which itself raises questions about the necessity of the "language" component.

**Questions:**

1. When the VLM generates the next waypoint $w_t$, does it condition on the previously generated waypoint $w_{t-1}$? The autoregressive formulation $p(w_t | w_0, ..., w_{t-1})$ (Eq. 5) implies this. If so, how is the continuous coordinate vector $w_{t-1}$ encoded and fed back into the language model, which typically expects discrete tokens? If not, why is the model described as "autoregressive"?
2. The paper mentions that using special tokens reduces computational overhead, but also lists VLM inference latency as a limitation. Is there a concrete performance gain in inference speed (e.g., FPS) compared to baseline models?

---

### Official Review · Reviewer_R4qf · 2025-10-31

**Soundness:** 2
**Presentation:** 2
**Contribution:** 2
**Rating:** 4
**Confidence:** 5

**Summary:**

The paper presents a framework, Max-V1, aimed at enhancing trajectory planning in autonomous driving by adapting a general-purpose Vision-Language Model (VLM). The authors propose a method that modifies the VLM’s output tokens to predict a set of 2D coordinates, effectively transforming a classification task into a regression task. Additionally, the framework integrates LiDAR data to create RGB-D inputs and applies different scaling factors to trajectory points based on their spatial regions. The authors claim that their approach achieves state-of-the-art performance on the nuScenes dataset, demonstrating improvements in displacement error metrics compared to previous baselines.

**Strengths:**

1. **Performance Improvement**: The paper reports a significant reduction in displacement error metrics, with an overall improvement of over 30% compared to prior baselines on the nuScenes dataset, indicating strong empirical performance.
2. **Integration of Multi-Modal Data**: The combination of RGB and LiDAR data for trajectory planning is a relevant approach, as it leverages the strengths of both modalities to enhance prediction accuracy.
3. **Statistical Modeling**: The authors provide a statistically grounded model for waypoint representation, which may offer a more principled foundation for loss function design in VLM-based driving research.

**Weaknesses:**

1. **Lack of Innovation**: The primary contribution of modifying the VLM’s output to predict 2D coordinates and the integration of multi-sensor data appears to be a straightforward application of existing techniques rather than a novel approach. The transformation of a classification task into a regression task has been widely adopted in various VLA methods, which diminishes the perceived innovation of this work.

2. **Insufficient Evidence of Scaling Factors**: The results presented in the DELFT dataset do not convincingly demonstrate a significant difference in performance when using scaling factors. This raises questions about the effectiveness and necessity of this approach.

3. **Potential Overfitting**: The training methodology, which involves using both the training and validation datasets, may lead to overfitting to the specific data distribution of the dataset. This limitation undermines the generalizability of the findings and does not convincingly support the effectiveness of the proposed strategy on other datasets.

4. **Limited Evaluation Scope**: The main results are primarily based on the nuScenes dataset, with no experiments conducted on other relevant datasets such as NAVSIM and Benchdrive. This lack of diverse evaluation limits the robustness of the claims made regarding the framework's performance and applicability.

**Questions:**

See above.

---

### Note · Authors · 2025-11-18

I have read and agree with the venue's withdrawal policy on behalf of myself and my co-authors.